# Mindfulness in the Context of Engaged Buddhism: A Case for Engaged Mindfulness

Brian D. Somers 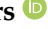

Department of Buddhist Studies, Dongguk University, Seoul 04620, Korea; briansomers29@dongguk.edu

**Abstract:** This article investigates mindfulness-based interventions (MBIs) in a clinical setting and considers the benefits of socially engaged mindfulness practices. The main aim is to consider the relationship between MBIs, especially as a clinical practice (including disengagement from negative ruminations and difficult emotions) and Buddhist mindfulness as a practice of social engagement for systemic change. While MBIs and engaged Buddhism both aspire to ease suffering for individuals and societies alike, they differ as the former emphasizes psychological treatment of the individual and the latter includes a call to action for more widespread change in the political, economic, and social arenas. At the center of this article is an inquiry into mindfulness practice in relation to engagement, disengagement, and re-engagement with objects of the internal and external world and what that means for the practitioner as well as society at large. It will be concluded that the amendment of mindfulness-based practices with lovingkindness and compassion-based practices shifts the emphasis from the clinical treatment of an individual patient toward a more holistic approach that includes the wellness of all beings. This shift is desirable and necessary as it considers a broader set of causes of psychological suffering and helps to reconcile the divide between disengaged cognitive practice and social engagement.

**Keywords:** mindfulness (念); mindfulness-based interventions (MBIs); non-judgmental awareness; discriminative awareness; socially engaged Buddhism; engaged mindfulness





## 1. Introduction

In the wake of the Second World War, America saw the rise of fringe groups and antiwar movements often mixed with the mysticism of the East, some of which first made its way to America in the late nineteenth century at the World's Parliament of Religions at the World's Fair in Chicago. At this time, Buddhist teachings were reaching new, foreign audiences and being blended with science, in part due to the work of Henry Steel Olcott, Paul Carus, and Anagarkika Dharmapala (McMahan 2008, p. 91). Interest in Buddhism continued to grow in the West and was made more widely available beginning in the first half of the 20th century by such teachers as D. T. Suzuki followed by Aldous Huxley, Erich Fromm, Robert Blyth, and Carl Jung, to name a few. Buddhism was further popularized by such figures as Jack Kerouac, Allen Ginsberg, Alan Watts, and composers such as Philip Glass and John Cage.

Having studied and practiced under Burmese meditation teacher Sayāgyi U Ba Khin (1899–1971), American aerospace and weapons engineer Robert Hover (1920–2008) played an important role in the establishment of mindfulness in America. After being appointed by U Ba Khin in the mid 1960s, Hover engaged in the implementation of "vipassanā-derived systems for social, economic, and political purposes" (Stuart 2017, p. 160). The shift from Asia to America, however, is a complex one in part due to the rejection of what could be perceived as mystical and unconventional methodology. Responding to the challenge to incorporate mindfulness meditation with American culture was molecular biologist, Jon Kabat-Zinn. Having been influenced by Hover, Kabat-Zinn would go on to create mindfulness-based stress reduction (MBSR), the fruit of his efforts to incorporate the teachings of mindfulness in what has been called

a process of the "epistemological masking of the possibility of religious experience" (Stuart 2017, p. 174). Science would be a primary means of cementing mindfulness in the West and MBSR marked its beginning by including mindfulness in a clinical setting. The success of the program would go on to initiate a surge in mindfulness-based practices sometimes referred to as, "mindfulness mania" (Buswell and Lopez 2014). Although some may still see mindfulness as a fad, others have referred to it as a revolution being hailed as "a small thing that can change the world" (Boyce 2011). Modern mindfulness has grown to be practiced not just in hospitals and during counseling sessions, it is being taught in schools, in the military, to the police and civil servants, and in the workplace favored especially by the tech industry, among others.

Given the surge in popularity of mindfulness over the past few decades, it has naturally been met with some skepticism including worries regarding the pathologizing of emotions (Purser 2019, p. 45), the depoliticization of suffering (Hyland 2017), as well as the gap between secularized techniques and Buddhist teachings (Thompson 2020, pp. 48–54). There are further concerns that consider some modern mindfulness practices as a watered-down version of the traditional, which leads to questions regarding colonization and appropriation of Buddhist teachings (Poceski 2020). Furthermore, the commodification of mindfulness teachings, which have in some cases been packaged and sold for remarkable profit, has also been a cause for concern. Therefore, some worry that modern mindfulness has been extracted from the Buddhist canons leaving integral ethical teachings behind (Brazier 2018). The relationship between traditional mindfulness and its modern cousins makes for a very rich ground for inquiry. While there are many elements to consider, this article focuses on socially engaged Buddhist mindfulness as it has come to rise in the second half of the 20th century and the relationship it has with modern mindfulness or mindfulness-based interventions (MBIs). The main aim is to consider the relationship between MBIs as a clinical practice of disengagement from negative rumination/difficult emotions and engaged Buddhism as a vehicle for systemic change. While both MBIs and engaged Buddhism have the same goal: to ease suffering, they differ as the former emphasizes psychological treatment of the individual and the latter includes a call to action for political, economic, and social change.

Before embarking on this investigation, it is first necessary to recognize the problem in lumping multiple mindfulness programs together under the single heading "mindfulness-based interventions". While mindfulness-based practices are common to all MBIs, there have been significant developments over the years. Following the innovative research of mindfulness-based stress reduction (MBSR), more recent, compassion-based programs have added a layer of lovingkindness to mindfulness practice, which highlights a warm integration of the insights that arise out of disengagement from the ruminating mind. This integration of lovingkindness is partially a response to mindfulness as an individualistic practice and thus includes compassion for all beings, including the self. The next section of this article begins by investigating the foundations of MBI practices, namely disengagement from a ruminating mind. This is done to contrast these practices with the engaged practices of socially engaged Buddhism. After considering engagement and disengagement as it pertains to mindfulness practices, compassion-based practices will be considered along with the role they play in social engagement.

## 2. Mindfulness and the Practice of Disengagement

Mindfulness-based programs include techniques which help the practitioner to disengage from the ruminative thoughts and emotions related to psychological suffering. These practices allow opportunities for thoughts to be seen as thoughts, not reality, therefore, setting the stage for more accurate and insightful engagement with phenomena of the mind and the objects in the world which they index. As Stahl and Goldstein (2010, p. 85) write in the MBSR workbook, "As you become aware of the

stories you spin and the traps you create, you can begin to disengage from them". This is better understood with an explanation of negative feedback loops, which is a particularly important element of mindfulness as it is taught in mindfulness-based cognitive therapy (MBCT). A negative feedback loop occurs when some output function of a system is fed back into the system in a manner that tends to reduce changes in the output. When applied in the context of psychology as negative thinking patterns, feedback loops are an example of ruminative thinking where an individual maintains a belief (regardless of its truth value) which leads to the perpetuation of a particular emotion or mood, often depression and/or anxiety.

Some three hundred and fifty years after Pascal distinguished between two types of minds (mathematical and intuitive or esprit géométrique and esprit de finesse) (Pascal 1941, pp. 3–6), Barnard and Teasdale developed interacting cognitive subsystems (ICS). ICS provides a detailed description of a systemic model of human cognition through the organization that underlies how humans process information, extract meaning, and control responses and actions. The fundamental processes include four primary subsystems which, among other functions, conceive of objects in a meaningful way and five peripheral subsystems, which roughly coincide with sensory perception and resulting reactions. Simply stated, ICS is centered upon a principal network consisting of two primary components known as the propositional and implicational subsystems. The propositional subsystem is originally defined as that which expresses semantic entities (Barnard and Teasdale 1991, p. 10). It functions conceptually, representing meanings that are either true or false (p. 23) and thus represents the cool rational thought of cognitive restructuring or "knowing with the head". The implicational subsystem is the other primary component of the principal network. It is informed by propositional sources (among others), which remains latent in meaning until developed through what is known as buffered transformation. Once the information is transformed, it takes the form of "higher-order meaning", also referred to in terms of "schematic mental models of experience" (Teasdale 1999, p. 61). These models represent patterns and themes that have been established from experience. Rather than knowing specific data as in the propositional subsystem, a more holistic, subjective sense of intuition or "knowing with the heart" is developed through the implicational subsystem.

It is the reciprocal relationship between the propositional and the implicational that constitutes what ICS calls the "central engine" of the mind. It is this cyclical relationship that mindfulness, especially MBCT, aims to intervene. By engaging in mindful-based techniques, feedback loops can be disengaged, and schemata adjusted to represent one's circumstances more accurately. According to the ICS framework, the two primary strategies to reduce the risk of returning depressed episodes are based on the adjustment of dysfunctional schematic mind models through a process of "identify and answer". First, individuals are encouraged to create alternative models to be accessed in situations that normally trigger a depressogenic model. Second, individuals learn the skills needed to disengage from the interlock cycles of the central engine. Through practical exercises, one begins by identifying the mode she is functioning within and responds by disengaging and shifting to a more present state of experience. These two steps are performed in the name of cognitive restructuring which aims to uproot the causes of depression at the schematic level, where habitual thoughts based on past experiences underlie dysfunctional assumptions.

When seeking to promote mental well-being, mindfulness programs take a similar "identify and answer" approach. Once identified, the shift from conceptualizing to mindful experience is often done by shifting from internally oriented to externally oriented attention. Sensations in the body are particularly useful to help disengage the feedback loops of the central engine and therefore meditations that emphasize sensations of the breath or body-scans are frequently used. Sometimes referred to as disengaging from autopilot (Teasdale et al. 2014, p. 41) or negative thinking patterns (p. 12), MBIs suggest a process where one distances himself from his thoughts with

the techniques laid out in the structure of the program, and through reflection and sharing with other participants, he gains insights which allow him to undermine the mental schema, which perpetuates depression, anxiety, and such. In other words, the mindfulness participant is taught to disengage from his thoughts which would otherwise hamper the cultivation of insight. Upon developing insight, the practitioner reengages, which is to say he engages in a new way that is more beneficial to psychological wellness.

In the case of MBCT, the first five sessions focus on disengaging from what is known as "doing mode" and shifting into "being mode". This act of disengaging is also known as "decentering" or "distancing" (Segal et al. 2013, p. 36), which enables the practitioner to relate to, not from, his thoughts and by so doing reset his depressogenic cognitive model. This is done instead of adjusting the content of the thought, which is already subsumed within a depressogenic model. The act of shifting is one from a first-person perspective where the thoughts are lived through, to a third-person perspective where thoughts are seen as events, not reality. By seeing thoughts as thoughts, or thoughts as mere events, the urgency to react is dampened and one learns to be curious, not reactionary, regarding their emotional alarm bells.

Based in part on the buffered mode of the above-mentioned propositional subsystem, the doing mode is described as a result-focused way of working with objects and goals. "The job of this mode of mind is to get things done—to achieve particular goals that the mind has set" (Segal et al. 2013, p. 68). This is done through a function of the mind referred to as the discrepancy monitor where one observes the current situation, the desired situation, and the gap between them that needs to be traversed. According to MBCT, the doing mode of mind is a very effective means of problem-solving external problems. However, when applied inwardly it strengthens the unwanted emotion. By applying the doing mode to the experience of depression, the depression becomes intensified. In line with ironic process theory, sometimes referred to as Dostoevsky's white bear problem (Dostoevsky 1997, p. 49), mindfulness-based programs assert that focusing on an object of the mind will strengthen it, even if the intention is to change or eradicate that object. Therefore, when working with difficult emotions one must approach from a non-analytic or non-judgmental perspective, what is referred to as being mode. Although difficult to define, being mode may be thought of as the opposite of doing mode in that it is not devoted to achieving goals. It is the state of mind where "feelings, sensations, and thought are directly sensed as aspects of subjective experience, rather than being the objects of conceptual thought" (Teasdale 1999, p. 68). Rather than looking ahead to what could or should be, and therefore being unsatisfied with what is, and rather than feeding the conceptualization of thoughts and feelings as real things, MBIs guide practitioners to anchor in the here and now accepting and allowing what is, without an urgent need for change. Ironically, such acceptance often leads to change. "The curious paradox is that when I accept myself as I am, then I change" (Rogers 1961, p. 17).

At first, mindfulness practices focus on teaching the difference between doing and being and how to shift from the former to the latter. With this emphasis, it is easy to overlook, especially for a beginner, that mindfulness practices do not intend to villainize the doing mode, but rather show how it can be detrimental to mental well-being, especially regarding depression. Therefore, by session seven of the eight-session MBCT course, consideration is given to doing and the importance of action. Although the doing mode of mind has been at the center of rumination and aversion, which often leads to depression, by the end of the program it is ensured that these practices are not anti-action and do not promote complacency. On the contrary, the shift from doing to being or from thought to experience helps to settle the mind so the practitioner has a clearer perspective from which to choose how to improve her well-being (and hopefully the well-being of others as well). Allowing or acceptance is "a vital *first step* to become fully aware of difficulties and to respond to them skillfully"



(my italics) ([Segal et al. 2013](), p. 293). In other words, awareness alone is not sufficient. Skillful navigation to prevent further depression is key. This is especially important as lethargy is a common struggle among those who face depression. Thus, session seven of MBCT advocates for pro-active changes, based on the insights brought about through mindful practices, to safeguard against future episodes of depression. "[I]n depression, we have *to do* something before we are able to want to do it" (my italics) (p. 343). Therefore, participants are encouraged to consider which activities give them a sense of well-being and to engage in them.

However, mindfulness meditation techniques are sometimes interpreted as a sufficient means to manage a wide variety of ailments. Mindfulness-based programs are often promoted as a self-help tool to manage problems such as ADHD, anxiety, depression, chronic pain, fatigue, anger, headaches, high blood pressure, and sleep problems ([Wong 2020]()). Taking things a step further, mindfulness meditation has also been presented as a way to achieve real-world goals such as being more productive, more effective at work, earning more money, etc. ([Tan 2012](), p. 235). Although these practices and achievements could lead to benefits for the world, a plan and guidelines for skillful action are not always described. While it is ultimately the responsibility of each practitioner to mind the hype and not conflate mindfulness-based programs with mindfulness for productivity and self-improvement, there is an overlap between the two. Furthermore, the relationship between Buddhist mindfulness as it is traditionally practiced, and its modern counterparts also needs to be considered. As beneficial as mindfulness techniques are, MBIs do not explicitly address the causes of psychological suffering that exist outside of the head.

As there is a diverse and unique set of conditions that affect each of us, MBIs are not a one size fits all treatment. MBIs teach that mindfulness practice is meant to cultivate insight and thus skilled actions and habits that are in line with well-being. However, disengagement may take the spotlight leaving rumination, and by extension, doing mode to be interpreted as the villain. Furthermore, the extent to which an MBI participant reengages with their environment requires deeper investigation which MBIs do not always provide. As a cognitive approach to easing symptoms of depression, anxiety, etc., the focus of the practice is the interior world of the individual. Therefore, one of the primary worries for some skeptics is mindfulness as a treatment of symptoms, but not a rectification of the causes of the symptoms. There is a concern that some mindfulness practices may be done in such a way that the status quo of society with grave inequalities, and therefore suffering, is maintained.

Therefore, some worry that MBI practices focusing on practices of "non-judgement" and "disengagement" may be misunderstood. While disengaging is a necessary practice for breaking the habit of rumination, could the practice mutate into a form of passivity, complacency, or even anti-activism? Depending on the program facilitator, the participants are not necessarily guided to distinguish between helpful and unhelpful discrimination. If one is not careful it is conceivable that thinking could be sidelined along with rumination. Therefore, a more detailed discussion about the shift from doing to being and *back into doing* is desirable.[1]

## 3. Engaged Buddhism

Buddhist masters have been carefully minding the shift between conventional to ultimate truths and implementing insights back into a convention world for centuries. Originally expressed by Tang Dynasty Chinese Chan Master Qingyuan Weixin (青原行思),

> In the beginning of our practice 'mountains are mountains and rivers are rivers.' They are manifested in the whole phenomenal universe. After much practice, the self is forgotten and 'mountains are not mountains and rivers are not rivers.' As the practice continues and matures even further, 'mountains are again mountains and rivers are again rivers.' The mountains and

rivers of the beginner are not the mountains and rivers of the mature prac-
titioner. However, the mountains and rivers of the mature practitioner are
identical with the mountains and rivers of the beginner. Do you see?

(Loori 2002, pp. 83–84)

The shift from mundane knowledge to supramundane wisdom is explained here
in three steps, the first of which is the nominal interpretation of phenomena. A label
and uttered sound are attached to an object for purposes of communication and con-
ceptualization. Next, the veil of convention is broached, and mountains are no longer
mountains, they are phenomena that one projects upon. Finally, there is a return to
the mountains as mountains, but this time they are perceived with the memory of a
new experience. Mountains are again mountains but in a new way.

For proponents of engaged Buddhism, the implementation of Buddhist teachings
in the world emphasizes social change. The history of engaged Buddhism generally
begins with the late Thich Nhat Hanh in the mid-1960s. However, slightly earlier
forms of engaged Buddhism, sometimes known as neo-Buddhism or Navayāna Bud-
dhism took place in India in the 1950s. At that time economist and statesman, B. R.
Ambedkar (1891–1956) had been campaigning for India's independence but is partic-
ularly well known for his social and political activism against caste discrimination. As
an outcast Hindu, Ambedkar fought for the rights of Dalits by renouncing Hinduism
and eventually converting to Buddhism. In the mid-1950s Ambedkar established the
Buddhist Society of India and converted to Buddhism along with approximately half
a million of his followers.

Considering other faiths such as Christianity, Islam, and Sikhism, Ambedkar
chose to convert to Buddhism as he found it the most fitting religious path for coun-
tering social injustice. With political and social activism at the heart of his motivation,
the Buddhist movement for Dalits was a reinterpretation of traditional Buddhism.
Key Buddhist concepts such as karma, no-self, nirvana, samsara, and even the four
noble truths had little role in Ambedkar's Buddhism (Deitrick 2009). Instead, Ambed-
kar's reinterpretation of Buddhism centered almost exclusively on the issue of class
struggle and the promotion of social equality.

Political and social justice as a theme in Buddhism continued. Less than a decade
later, Thich Nhat Hanh coined the term "socially engaged Buddhism" while promot-
ing peace activism, largely in reaction to the Vietnam War. At this time, he founded
the Order of Interbeing (Tiep Hien), the agenda of which he would come to define in
fourteen guidelines based on the extension of Buddhist practice for individual suffer-
ing to include the world at large. Hanh set "collective as well as individual liberation
as a soteriological goal" (Gleig 2021). Although engaged Buddhism primarily began
as a reaction to social injustice and war, it later grew to include a broader set of aims
related to "peace and non-violence, human rights, just and equitable development,
liberation from oppressive government, social and economic justice, prison reform,
access to education and health care, environmental protection and sustainability, and
gender and racial equality" (Edelglass 2009, p. 420).

Engaged Buddhism references the fourth of the four noble truths as a call to
action where one must work to overcome the suffering described in the first three
truths. Thus, the grand purpose of individual practices is emphasized, citing ethical
teachings as a guide to cultivate such qualities as generosity, moral discipline, pa-
tience, compassion, loving-kindness, abstaining from harming others, the monastic
code (Vinaya), right livelihood, skillful means in alleviating suffering, and the bod-
hisattva ideal for saving all sentient beings from samsara (Edelglass 2009, p. 420).
Based on the teachings of the Buddha as ethical practices to benefit all sentient beings,
Hanh derives the following guidelines.

*Openness* refers primarily to open-mindedness, especially regarding one's beliefs,
including religious beliefs. "The Buddha's teachings are a means of helping people.
They are not an end in themselves, they are not something to worship or fight over"

(Hanh 2020). Further commentary on this guiding principle warns against idolization and reminds readers of the analogy of the Buddhist teachings as a raft. Once one has reached the other side of the river, they are meant to discard the raft (teachings), not cling to them.

*Nonattachment to views* continues with the theme of non-attachment indicating the importance of opening the mind to include the points of view of others and therefore to recognize the impermanent nature of absolute ideas. "This training warns us not to get caught in our own knowledge or views" (Hanh 2020). By clinging to a particular point of view one sets an obstacle between herself and insight.

*Freedom of thought* teaches that not only can clinging to one's own point of view prevent him from insight, forcing others to abide by a particular point of view is also undesirable. Everyone must be allowed the opportunity to inquire into the points of view he may wish to follow. Regardless of tradition or sponsorship, each is encouraged to decide for himself whether the view in question causes harm.

*Awareness of suffering*, the fourth training, guides practitioners to be mindful of suffering itself and not to cover it up with distractions. "Suffering is not all bad. It can have a therapeutic power. It can help us open our eyes. Once we start facing the suffering within us we shall want to search for its cause [...] We also want to find out the causes of the suffering in our society" (Hanh 2020). One of the primary reasons for being with suffering is to transform that suffering into compassion, as well as peace and joy.

*Compassionate healthy living* is the fifth guideline which emphasizes simple living for the sake of limiting distractions through overconsumption (which includes food, sense impressions, volitions, and consciousness), as well as to limit excessive class and wealth division. Fame, power, and wealth are seen as obstacles to a happy and healthy society and so, practitioners are encouraged to consume less, live simply, and "remain as free as possible from the destructive momentum of the social and economic machine, to avoid modern diseases such as life stress, depression, high blood pressure, and heart disease" (Hanh 1987, p. 37).

*Taking care of anger* is an important practice as anger blocks effective communication. When anger arises, rather than being sucked into the experience and therefore cultivating and reenforcing the emotion, it is suggested that one shift into the practice of mindfulness, anchoring in the breath or bodily sensations instead of the anger. This can be done through the cultivation of right diligence.

*Dwelling happily in the present moment* is the seventh guideline, the importance of which is the emphasis on inner experience as practitioners are reminded about the power of interpretation. While external conditions influence our state of well-being, ultimately, happiness depends on our mental attitude. Rather than worrying about the future or regretting the past, recognizing the present as the only point at which life is available is a powerful means of cultivating well-being. This is done by learning to see the healing elements that exist in all situations.

*Community and communication* highlight collective insight. The core of cultivating collective insight is compassionate communication, which is inclusive of all beings. The practices included in the three harmonies of speech, thinking, and views are the basis for developing this collective insight. Hanh reminds us "[t]o reconcile is not to judge by standing outside of a conflict. It is to take some responsibility for the existence of the conflict and to make every effort to understand the suffering of both sides" (Hanh 2020).

*Truthful and loving speech* recognizes the power of words as a cause of karmic results. Just as deeds or actions have a direct effect on the conditions of the world so too does speech. Therefore, practitioners are reminded to use words skillfully for wholesome ends. This can be done by speaking the truth, not exaggerating, not speaking divisively, and not using abusive speech. This includes deep listening. Listening deeply allows us to understand another and understanding another sets the stage for compassionate speech.

*Protecting and nourishing the sangha* refers not just to the brother/sisterhood of monks and nuns but to the global community at large. Such a community is described as a vast colony of individual cells which must be synchronized with each other for the overall health of the organism. Thus, rectifying problems must be done in a non-partisan way while a clear stand against injustice is maintained.

*Training right livelihood* reminds us to behave responsibly as citizens and consumers. One must be thoughtful when choosing her way of life, especially her vocation. This is largely a comment regarding our global-economic system where sponsorship and support can be given remotely through investments of time and money. One need not remove herself from the system entirely, but her consideration regarding contributions to the health of the planet through habits of consumption and sustenance is essential.

*Reverence for life* refers to the dedication necessary to protect life. It emphasizes antiwar sentiments through the promotion of peace education, mindfulness meditation, and reconciliation amongst the divided. This applies not only to human beings but to all living creatures from mammals to microscopic organisms and everything in between.

*Generosity* is the thirteenth guideline which urges us to be aware of social inequality and promote responsible earning and sharing. While there is nothing wrong with earning resources, generosity reminds us to earn in responsible and respectful ways. This includes not profiting from the suffering of others and working for the benefit of others.

*Training true love* is the final guideline which pertains to lay practitioners. This guideline highlights the difference between sexual desire and true love. The main point here is to apply loving-kindness, compassion, joy, and equanimity within the sphere of all relationships, sexual relationships included. The cultivation of interbeing is diminished in relationships that involve sexual misconduct and so the laity is guided to treat sexual connection as part of a long-term plan which strengthens a relationship as opposed to short-term gratification which intensifies isolation.

In his work *Responding to the Cries of the World: Socially Engaged Buddhism in North America*, Rothberg considers the distinction between engaged Buddhism and Buddhism itself. Considering the work of Thich Nhat Hanh, he suggests there are two main types of engaged Buddhism: engaged Buddhism for social change and engaged Buddhism for daily life. In other words, engaged Buddhism is not solely focused on large-scale political and social change; it also engages small-scale communities as well. It has been surmised that Thich Nhat Hanh considered engaged Buddhism to constitute three elements including, "Buddhism for everybody", "going into the world", and finally "getting involved" (Rothberg 1998, p. 273). These three elements grow successively as practice focused on cultivating awareness in daily life grows to serve society, especially regarding education and charity work. It can then develop even further through explicit activism for political and social change.

Engaged Buddhism has been considered not only as a means to face social problems, but as a part of daily life including the wellness of the individual. As Thich Nhat Hanh points out, many people, especially in the West, have found Buddhism from an intellectual perspective while searching for a means to ease their own psychological suffering. Used daily, engaged Buddhism in the form of meditation practice also serves as "a way of helping us stay in society" (Hanh 2005, p. 53) and therefore promotes individual wellness by reducing depression causing isolation. As a two-tier practice, engaged Buddhism for daily life sets the stage for large scale changes at the level of institutions. Insofar as Buddhism aims to ease the suffering of all beings, Buddhism is socially engaged Buddhism. As Rothberg says, "Buddhists who wholeheartedly practice 'in the world' do not call themselves 'engaged' Buddhist" (Rothberg 1998, p. 273).

Considering the expansion of Buddhist practices and teachings in the West, Prebish points out that there have been several development issues in American Buddhism, one of which is engagement (Prebish 1999, pp. 81–85; Gleig 2019, pp. 39–40). Especially in recent years, the general tone of engaged Buddhism has become more political and tends to remain as a categorical subset of Buddhism. While associated with progressive and liberal philosophies, engaged Buddhism has stood for easing suffering on a global scale, the aim of which is to liberate all sentient beings. In 1998, Rothberg writes, "[a]t this point in the evolution of North American Buddhism [...] there are no prominent politically 'conservative' Buddhist public voices and movements, and no clear criticisms of 'left-wing' socially engaged Buddhism ..." (Rothberg 1998, p. 271). However, this is no longer the case as right-wing and alt-right Buddhism has emerged in recent years. The politically conservative, right-wing agendas which oppose Buddhism as a left-wing movement often claim an apolitical stance while charging against the philosophies of "liberal Buddhists" (Gleig and Artinger 2021). Some such right-wing cohorts "believe that many facets of engaged Buddhism such as support for the Black Lives Matter movement, gender egalitarianism, and LGBTQ+ inclusivity are external and in opposition to the fundamental teachings of the Dharma". As Gleig and Artinger further point out, alt-right Buddhists "conflate liberal Buddhism with engaged Buddhism–seeing social justice efforts and dignity, equality, and inclusion (DEI) initiatives as essentially 'liberal' in nature" (Gleig and Artinger 2021, p. 32).

## 4. Criticisms of MBIs

While mindfulness meditation is evidence-based and has a great deal of research supporting it, some have pointed out the hype that has in some cases led to misinformation. As a result, some dubious claims regarding the effectiveness of mindfulness have been discovered (Van Dam et al. 2018). While some report that mindfulness is not as effective as claimed, others have noted that it may even be harmful if it reinforces previously held (harmful) beliefs. For instance, in a case where one has several false beliefs, some of which are consistently present and some which are not, the consistently present false beliefs may be reinforced (Moore 2016). Furthermore, the main issue taken up in a 2013 article in the Huff Post by Ronald Purser and David Loy is concerned with the social implications of mindfulness such as the commodification of religious practices (Purser and Loy 2013). Referred to as a capitalist spirituality, Purser elaborates on the original article in his 2019 book *McMindfulness*, expressing concerns that, contrary to its claim, modern mindfulness and the wellness industry may contribute to suffering insofar as it provides people with a means to cope with political, social, and economic causes of suffering, treating the individual symptoms and thus enabling the neoliberal agenda to continue.

In response to mindfulness being described as revolutionary, Purser writes, "Anything that offers success in our unjust society without trying to change it is not revolutionary—it just helps people cope" (Purser 2019, p. 7). Several pages later he continues,

> The so-called mindfulness revolution meekly accepts the dictates of the marketplace. Guided by a therapeutic ethos aimed at enhancing the mental and emotional resilience of individuals, it endorses neoliberal assumptions that everyone is free to choose their responses, manage negative emotions, and "flourish" through various modes of self-care. Framing what they offer in this way, most teachers of mindfulness rule out a curriculum that critically engages with causes of suffering in the structures of power and economic systems of capitalist society.
>
> <div align="right">(p. 11)</div>

Furthermore, the problem for Purser is not so much that mindfulness makes one disengaged or complacent. On the contrary, mindfulness is an engaged and active pursuit. However, this pursuit can be an individualistic one with coping mechanisms for

the practitioner as its aim. Thus, the concern is that mindfulness engages people in a way that allows them to feel productive and free while being curbed by an underlying economic system. Therefore, the criticism is of mindfulness as socially disengaged.

This interpretation of mindfulness is based in part on the premises that mindfulness can be practiced as an individualist project to develop personal wellness and that mindfulness at least in some cases is a passive practice insofar as engaged, discriminating thoughts are suspended when one disengages from the inward-turned doing mode (also known as the driven-doing mode) and fosters non-judgmental observation in the present moment. The latter of the two is of particular concern for this article and will be investigated further, below.

*Indiscriminate and Non-Judgemental Observation*

One of the key techniques of mindfulness meditation which help to improve mood is metacognitive practices to prevent negative rumination. In the case of depression, once an episode has commenced, it is much more difficult to slow the momentum of active thoughts than it is to prevent them from starting. Given that some cases of depression known as dysthymia persist for years (American Psychiatric Association 2013, p. 168), erroring on the side of prudence and disengaging from all thoughts that could trigger such a spell seems reasonable. However, despite the risks that accompany rumination, it is necessary to simultaneously acknowledge that critical, analytic, discriminative, or judgmental thinking is not necessarily problematic and often required. Despite the definition of mindfulness as non-judgmental (Kabat-Zinn 2013, p. xxvii), investigation is a crucial component of meditation practice. In the context of mindfulness, the term "investigation" or "analysis" is usually substituted with such terms as "curiosity" (Segal et al. 2013, p. 91) or "observation", where judgement is suspended for the time being to practice mindful techniques and observe what is happening (Kabat-Zinn 2013, p. lxv). This is a proficient way of deemphasizing the rigor and perhaps strained calculation that might otherwise accompany analysis. Despite the acknowledgment that critical thinking is not a villain, it can be conflated with negative rumination. To help make this distinction clearer, MBCT distinguishes between doing mode, which is described as a useful problem-solving tool and driven-doing mode, which is described as the application of the doing mode to problems of the mind. This is a useful distinction in the early stages of practice when one may be overwhelmed with thoughts and emotions. In a whirlwind of negative rumination, simple disengaging is necessary until the mind has settled. Modern mindfulness cultivates being mode described as the mode where "feelings, sensations, and thought are directly sensed as aspects of subjective experience, rather than being the objects of conceptual thought" (Teasdale 1999, p. 68). While this suffices an introduction to mindful mediation, especially in a clinical environment, it may leave a more seasoned practitioner considering practice beyond the meditation cushion. Therefore, a clearer distinction between negative rumination and deliberate thought is necessary for extended practice especially as it is incorporated into daily life.

As a long-term preventative measure for depression, MBCT is described as an active practice. Not to be mistaken for resignation, MBCT practice is "an active, willing gesture of acceptance and openness to experience. It takes conscious commitment and energy. In allowing/letting be we choose how to respond" (Teasdale et al. 2014). The practitioner is re-establishing cognitive patterns and habits so triggers of depressogenic schemata are not engaged. However, there are causes of depression which go beyond the cognitive that require active and conceptualized investigation. These include the nature of self and other, the philosophical investigation of which can lead to reducing the three causes of suffering (三毒) according to Buddhism: greed (貪), anger (瞋), and ignorance (癡). They also include exogenous causes of suffering such as social and economic inequality and repression, among others.

Intentional and deliberate action on the social level is required, and plans need to be made. Mindfulness courses equip people with the tools needed for action but not a guideline for what actions to take. In other words, mindfulness as a secular practice struggles to reconcile with religious values (Stanley et al. 2018, pp. 13–16). Thus, mindfulness-based courses could benefit from a more judgement-inclusive second phase, which places emphasis on active changes in the environment in addition to the restructuring of mind schemata. Insofar as the definition of mindfulness as non-judgmental is exclusively adhered to, mindfulness-based programs disproportionately emphasize acceptance without due consideration for the role active evaluation plays in traditional mindfulness practices. Some critics have gone so far as to describe mindfulness-based programs as the antithesis of critical thinking in that "our thoughts are [considered by MBIs as] the culprit every single time! ... critical thinking is pathologized in mindfulness" (Purser 2019, p. 45). The nature of mindfulness is also considered by Dreyfus who notes the evaluative nature of Milinda's questions in the well-known Buddhist analogy. This text is an example of "a description of mindfulness as being explicitly evaluative ...". The ethical dimensions of mindfulness are pointed out, which shows that "the function of mindfulness is not just to keep in touch with whatever is present in the ken of attention but also includes the not drifting away from the wholesome and unwholesome mental states" (Dreyfus 2011, p. 45). While this shows that Buddhist mindfulness is not exclusively non-judgmental, it is also another example of ethical considerations falling by the wayside in secular programs while emphasis is placed on techniques for self-improvement.

**5. In Defense of MBIs**

There are several reasons why MBIs emphasize mindfulness practice as an individual practice of non-judgement. First, MBIs are meant as a practice-based course where practitioners are invited to recognize and interact with their thoughts and emotions in a safe environment. These practices are based on bodily sensations to anchor the mind and foster being mode. Therefore, introducing thought-provoking, philosophical queries is, to some degree, at odds with the agenda to disengage from negative rumination. While traditional mindfulness practices are not non-judgmental, there is undoubtedly an underlying theme of calming and observing the mind (止觀). Moreover, the emptiness of self-nature (自性) is at the core of traditional mindfulness practices. However, emptiness (空) insofar as it describes no-self (無我) and impermanence (無常) can be easily misinterpreted. The term "emptiness" is itself frequently used to describe how one feels in a state of depression. Given that depression is one of the primary aims of MBIs, confirming that all phenomena, including ourselves, are empty could easily intensify despair. Emptiness has regularly been interpreted as nihilistic (Matilal 2002, p. 204). Furthermore, even if conceptually understood, emptiness and associated concepts are very challenging topics. Not all participants, especially in an introductory course, will be ready to face the connection between emptiness and self and therefore one's own mortality. These can be intense practices that might be best approached through gradual rather than sudden exposure.

Finally, contemporary mindfulness has been established in empirical research where it has been developed on evidence-based data. While scientific evidence garnered from a logically positivistic perspective is indispensable, an over-emphasis on quantitative data may overlook insights that arise through humanistic or holistic means. These include firsthand experiences such as feelings and intuition that accompany spiritual practices. While the careful mindfulness program facilitator will avoid going too deeply into conceptual inquiries that provoke negative ruination and thus interrupt the meditative experience, they are also cautious not to extend beyond the secular domain of religion or metaphysics. MBIs apply Buddhist-based practices insofar as they are scientifically justifiable, despite not engaging in a metaphysical in-

vestigation of teachings that support the techniques. Although they include spiritual-based practices, MBIs maintain a secular position.

The issue investigated in this article concerns MBIs and the degree to which they are engaged, especially regarding social engagement. Moreover, the inclusion of social engagement in mindfulness-based programs has developed out of compassion-based mindfulness. A case in point is Mindful Self-Compassion (MSC), which begins with mindfulness and considerations for the self but includes compassion-based practices (for both self and others) as the program progresses. Although the term self-compassion has sometimes been mistaken for a self-centered or selfish practice, the distinction between self-compassion and self-centeredness is made clear early in the course. "Some worry that by being self-compassionate rather than just focusing on being compassionate to others, they will become self-centered or selfish. However, giving compassion to ourselves actually enables us to give more to others in relationships" (Neff and Germer 2018, p. 20). Using the analogy of emergency procedures on an airplane, MSC explains that self-compassion is like affixing one's own oxygen mask first before assisting others (p. 139).

MSC is not the only example of mindfulness-based programs which go beyond practice for self-development, MBSR has also made an overt inclusion of others in MBSR 2. One of the key themes of MBSR 2 is "our relationship with the environment." This program "emphasizes kindness and compassion" and teaches that "responding wisely to choices in our lives means considering, and engaging with, our internal, individual, social, global and natural environments" (Brown School of Public Health 2019). Despite the emphasis to look inside the self to develop desirable qualities and reduce undesirable qualities, compassion-based programs also turn outward to include the environment and others. To continue these other-inclusive practices for wellness, it will be helpful to reflect on their traditional roots, which are inextricably intertwined with value ethics. While modern mindfulness is a secular practice it could find further support from the Buddhist teachings which have influenced it. This support could garner not only a more philosophically robust theory but also an even more effective means of easing suffering for all, including the self.

## 6. Engaged Mindfulness

Since the development of engaged Buddhism over fifty years ago, the teachings, especially of Thich Nhat Hanh, have gone on to influence practitioners today. Parallax Publishing was a major contributor to this cause as it continues to publish reading material focused on the engagement of Buddhist practices and teachings. Among the most well-known institutes focusing on engaged Buddhism are the Buddhist Peace Fellowship (BPF) co-founded by Nelson Foster, Robert Aitken Roshi, and Anne Aitken in 1978 and the International Network of Engaged Buddhist (INEB) established in Thailand by Sulak Sivaraksa in 1987. In a word, these networks can be described as inclusive. Although founded on Buddhist values, interconnection among Buddhists as well as non-Buddhists is fundamental. Motivated by social concerns, members aim to heal systemic harms. These networks are meant to serve as a hub for individuals to come together and support one another in their Dharma practices. In the case of INEB, the facilitated activities tend to include conferences, peacebuilding and reconciliation, human rights and social justice, alternative education, gender and womens' empowerment, Buddhist economics, alternative development, environment and climate change, reform/revival of Buddhist institutions, youth and spiritual leadership development, Buddhist art, and inter-religious/faith dialogues and collaboration (International Network of Engaged Buddhist 2022).

More recently established, there are several institutes which overtly include mindfulness as well as compassion, applying it in a way that permeates from the practitioner out into her environment. Already mentioned above is MSC in the Center for Mindful Self-Compassion (CMSC). In addition to this is the Center for Contem-

plative Science and Compassion-Based Ethics (CCSCBE) which promotes an ethical approach to mindfulness through a course in cognitive-based compassion training (CBCT). CBCT is described as a set of reflective exercises that "seek critical insights into the way one's mindsets and attitudes can be shifted to support personal resiliency, to foster an inclusive and more accurate understanding of others, and ultimately to intensify altruistic motivation" (Emery University Center for Contemplative Science and Compassion-Based Ethics 2022). Furthermore, the explicitly prosocial Engaged Mindfulness Institute (EMI) offers programs in mindfulness where participants have gone on to be "engaged in a wide array of community development, social change, and peacemaking activities." Some of these activities include: "conflict resolution, homeless advocacy, restorative justice, prison ministry, juvenile justice and at-risk you programs, AIDS relief, hospice and palliative care, social entrepreneurship, and human rights and peace work all around the world" (Engaged Mindfulness Institute 2020).

Mindfulness as a Buddhist tradition is based on ethical teachings centered on the principle of interdependent origination (緣起) also known as dependent arising or interbeing. Insofar as the well-being of the other fosters the well-being of the individual, programs utilizing mindfulness solely as a self-help technique not only challenge the Buddhist mindfulness outline but may also undermine their own agenda to reduce suffering. Mindfulness as a technique for alleviating depression, anxiety, etc. is a reasonable approach provided self-help is a first step toward a commitment to the well-being of all beings. The primary tension between mindfulness and its traditional, compassion-inclusive predecessor becomes clearer when investigating the difference between the secular and spiritual. As an evidence-based program, there may be a reluctance to engage with metaphysical or religious concepts. However, Buddhist practice is indeed a religious practice despite programs which highlight scientific elements. Buddhism is inseparable from moral judgements and the overall aim to awaken to ultimate truth (眞諦). As Thompson points out, "Buddhist theories of mind lose their point if they're extracted from the Buddhist normative and soteriological frameworks" (Thompson 2020, p. 13). While Buddhism includes both the secular and the spiritual, as described by the two truths (二諦) MBIs are based on the secular, which selects and incorporates certain spiritual practices insofar as they are evidence-based regarding specific clinical questions.

Insofar as modern mindfulness has been described as spiritual, some have noted that it constitutes a re-enchantment with nature, taking influence from the Transcendentalist romanticization of the wilderness. Thoreau placed "a high spiritual value on the solitary contemplation of nature" (McMahan 2008, p. 167) and Kabat-Zinn follows suit, writing, "Henry David Thoreau's two years at Walden Pond were above all a personal experiment in mindfulness" (Kabat-Zinn 2005, p. 24). As such, mindfulness has been referred to as a repackaging of "Thoreau's celebration of nature and aimless strolling" (Purser 2019, p. 228). Transcendentalists such as Thoreau who delve into the romanticism of nature and often secluded themselves from society have unquestionably influenced Kabat-Zinn and therefore the roots of mindfulness-based programs. However, to say that the Transcendentalists and Kabat-Zinn were moved by their private experiences with nature is not to say they were socially disengaged or complacent. Before publishing his journals written in his cabin on Walden Pond, Thoreau wrote *Civil Disobedience*. Here, Thoreau uses the word "civil", not as a synonym for "polite", but with reference to concern for citizens. *Civil Disobedience* is a manifesto calling for resistance to injustices by the state. The first line of this essay states, "That government is best which governs least" (Thoreau 2012, p. 275), and so Thoreau was not only promoting individual freedoms; he was in opposition to government in general. Therefore, insofar as he is a fierce supporter of individualism, he is an excellent role model for mindfulness-based practices. However, this right to freedom was meant for all individuals and thus *Civil Disobedience* can be read as a call to action against a government that at that time supported slavery and was using tax dollars to fund the

Mexican American War. Although an individualist and one who took great efforts to seclude himself from the torments of modern social living, his motivations for living such a life seem to be based on his yearning for equality and justice. His "aimless" strolling in nature was founded on a call for systemic change, not a way to find personal peace of mind in an unjust state. The peace of mind Thoreau had was not merely influenced by his close relationship with nature but because solitude in nature served as his home base from which he was able to engage with social injustice.

Modern mindfulness programs remind us of the home base where one can find solace. Like Thoreau, Kabat-Zinn's work guides participants to stabilize their ruminating minds by getting in touch with nature and practicing breathing exercises and concentration techniques. MBSR among other mindfulness-based programs has helped many to settle their minds in a world that includes much suffering and turmoil. However, if the world itself is sick, modern mindfulness is only addressing the first half of the problem. Mindfulness as described by Thich Nhat Hanh in the context of interbeing is socially engaged mindfulness, which finds its roots in the Vinaya (律) or precepts of Buddhist teachings. "Mindfulness must be engaged. Once there is seeing, there must be acting. Otherwise, what is the use of seeing?" (Hanh 1992, p. 91). Secular mindfulness supported by similar guidelines could help to ensure that 8-week mindfulness programs, which are sometimes used as a set of techniques for personal well-being, include social well-being as well. The shift to ensure the inclusion of others has already begun in the form of compassion-based mindfulness programs and continues to become even more engaged with Buddhist Derived Practices (Van Gordon et al. 2018, p. 261). That said the distinction between religious and secular meditation practices remains blurred. As a result, terms which encompass both the religious and secular are used. Such terms include spirituality, spiritual-based practice, ancient or traditional practice, contemplative practice, etc. One of the primary differences between the two seems to be individual freedom regarding the resistance to following prescribed practices on faith. Regarding modern mindfulness, a treaty seems to have been formed between religion and science where an "it's okay as long as it reduces suffering" attitude has been generally accepted.

## 7. Conclusions

Initially, MBIs are at odds with activism and even discriminative thinking, but this is a necessary first step for most people. The challenge currently faced is what to do after completing a mindfulness-based course. It is necessary, from the beginning, to emphasize that mindfulness is not a one-size-fits-all or universal fix. It is a self-centered starting point, which then expands to include others. The question regarding mindfulness and Buddhist engagement is not whether one is engaged but to what extent (and how) one is engaged. For mindfulness as a modern therapeutic practice and for mindfulness as a traditional Buddhist practice, the difference resides with the intention of the practitioner. While the former aims to ease psychological suffering, the latter aims to liberate all beings from the causes of suffering: anger, greed, and ignorance. Modern mindfulness has been adapted from its Buddhist roots (Helderman 2019), leaving behind specific Buddhist values. However, as it grows, it implicitly abides by certain, albeit general, guidelines. For example, MSC assumes that easing suffering for all is good when practicing lovingkindness which includes the development of four phrases for the benefit of self and other. These phrases include "may you/we/I be happy," " . . . be peaceful," " . . . be healthy," " . . . be with ease" (Neff and Germer 2018, p. 66).

While the way one thinks influences the way one feels, for the mindfulness revolution there is a disproportionate emphasis on the inner experiences of the individual, sometimes neglecting the individual's activities in the world. This leads to the consideration of what appropriate action is (consider Hanh's eleventh guideline, training right livelihood). As was shown in section three above, Thich Nhat Hanh gives a

fourteen-point guideline which defines just that. Modern mindfulness gives no such guidelines regarding value judgements or ethical behavior. Therefore, insofar as MBIs aim to ease individual suffering by disengaging the harmful cognitive schemata and not judging the content of experiences they could be setting the stage for disengagement as social inaction. However, depending on the direction of the practice and the capacity of the teacher, MBIs could instead be used as a tool to make wholesome judgements and develop compassion and understanding for those seen as opponents. Therefore, answering the question "Are MBIs socially engaged?" depends on the overall trajectory of the practice and the intention of the individual. As a fix that works to cause one to feel better through skillful coping, no. As a practice to make one more insightful and a better thinker (not a ruminator), yes.

Like acetaminophen, mindfulness that works when applied to ease symptoms is a technique that is helpful and can be used to ease suffering. However, so long as the external stressors remain, the headache returns. Like the Dharma, mindfulness that is practiced in accordance with faith and direct experience helps to reveal insights into the conditions of reality which cause suffering. This is a way of practice that eases suffering not by treating it but by understanding and then acting accordingly. However, the two are not mutually exclusive, which is to say that the former can be a necessary first step which leads to the latter. Before one can engage in social action, it may be necessary to disengage and reengage in a more insightful way.

**Funding:** This research received no external funding.

**Institutional Review Board Statement:** Not applicable.

**Informed Consent Statement:** Not applicable.

**Conflicts of Interest:** The author declares no conflict of interest.

## Note

[1]　This article argues in favor of an engaged form of mindfulness where mindfulness-based practices begin with disengagement from thought. For an indepth discussion on the details on the mindfulenss-based practices themselves and how they relate to traditional Buddhist teachings see Somers and Song (2021).

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
