# Peer review of "Mindfulness in the Context of Engaged Buddhism: A Case for Engaged Mindfulness"

_religions, doi:10.3390/rel13080746_

Round 1

Reviewer 1 Report

The present manuscript presents an interesting discussion – although one that has been had before. To make this paper unique, a more detailed and critical discussion would be necessary. Please see more detailed comments below:

-I am not sure why this paper is referred to as “thesis”. Was this part of a postgraduate thesis? I suggest re-wording it.

-Page 4: provide references to back up the claims about pathologizing, depoliticization, gap between secularized and Buddhist practice, and commodification of mindfulness teachings. Overall, throughout the manuscript, more claims need to be backed up with references. That way, the manuscript will make a useful contribution to the field. For example, Stanley, Purser, and Singh published a book on ethical issues in mindfulness, where some of the ideas mentioned here had also been raised.

-The contribution of the manuscript would be enhanced if it actually analyzed the content of programs such as MBSR and illustrated how some content is not represented there. This way, you are not only relying on some broad general references but have some concrete discussion points.

Author Response

Thank you kindly for taking the time to read my manuscript. I have carefully read and considered your comments and have made the following changes.

Firstly, I have taken your advice to change the term “thesis” and replaced it with “article”.

Next, your suggestion to include better referencing regarding pathologizing, depoliticization, the gap between secularized and Buddhist practice, and the commodification of mindfulness teachings has been well received. I have indeed added references including two articles from the suggested text.

Certainly, a more substantial account could be given if a mindfulness-based program were analyzed in detail. Please note that I have addressed this in a footnote on line 247.

With your constructive comments in mind two new sections were also included to this article (see lines 397-437 and 601-618) along with more thorough referencing – thirteen new references in total.

Thank you.

Reviewer 2 Report

This article is well written and interesting from Buddhist, mindfulness, psychological, ethical and activist perspectives.  It seems that 11 of the 14 pages are dedicated to MBIs, etc., therefore it lacks balance as it relates to research on socially-engaged Buddhism.  The abstract reads:

This thesis investigates mindfulness-based practices in the context of socially engaged Buddhism.

In the article, the balance is towards the context of clinical, not socially engaged.  The article could be improved with more attention to mindfulness in social engagement.

Author Response

Thank you for taking the time to read and consider my manuscript. I appreciate your constructive comments and have taken them into serious consideration. As a result, I have made the following changes.

As the reviewer rightly points out, given that this paper does indeed emphasize MBIs despite engaged Buddhism being an integral element, two sections have been included to expand the breadth of research on socially engaged Buddhism (see lines 397-437 and 601-618).

Furthermore, a minor change has been made to the abstract to reflect the content of the article more appropriately.

Finally, a total of thirteen new references have been included to support the content of the article more thoroughly.

Thank you.

Reviewer 3 Report

Dear authors,

Thanks for offering such a rich and well informed analysis on MBI's in the context of Engaged Buddhism.

The combination of advocacy and critical analysis is rare, and well appreciated. You have great knowledge of the complexities of how mindfulness practices arose in Western culture, can refer back to historical concepts, by also apply it to today's challenges as well as the fast emerging field of engaged Buddhism.

I applaud this contribution and have no further remarks.

Author Response

Thank you for taking the time to read my manuscript. I very much appreciate your kind support.

Round 2

Reviewer 1 Report

Thank you very much for the revisions.

Author Response

Thank you for your feedback. Much appreciated.